# Metagenomic Analysis of Liquor Starter Culture Revealed Beneficial Microbes’ Presence

**DOI:** 10.3390/foods12010025

**Published:** 2022-12-21

**Authors:** Ahmad Ud Din, Waqar Ahmad, Taj Malook Khan, Jun Wang, Jianbo Wu

**Affiliations:** 1Drug Discovery Research Center, Southwest Medical University, Luzhou 646000, China; 2Luzhou Ronghe Weizhen Biotechnology Co., Ltd., Luzhou 646000, China

**Keywords:** Zaopei Baijiu Daqu, metagenomics, liquor starter culture, beneficial microbes

## Abstract

Wines are complex matrices famous for their pleasant aroma and exceptional flavor. Baijiu (white wine) is a traditional Chinese liquor with a soft mouthfeel, fragrant taste, and long-lasting aftertaste. Baijiu is distilled from sorghum and wheat via solid fermentation. As in wines, the microbial ecosystem of Baijiu is a key decisive factor influencing aroma and consumer preferences. Microbial diversity in Baijiu has been intensively investigated. It is important to note that probiotics are a mixture of bacteria and yeast primarily intended to improve health. Our study aimed to characterize the microbial ecosystem of Zaopei Baijiu Daqu (ZBD) starter cultures for specific microbes with probiotic properties. The DNA samples of ZBD starters were analyzed using a metagenomic 16S rRNA approach to characterize the bacterial and ITS for fungal diversity. *Weissella cibaria* was the most dominant species in the bacterial community, while *Saccharomycopsis fibuligera* was the most abundant fungal species. Furthermore, functional prediction analysis identified unique pathways associated with microbial diversity relevant to functional innovation. These associated pathways include fermentation, amino acid metabolism, carbohydrate metabolism, energy metabolism, and membrane transport. This study identified beneficial microbes in the starter culture, opening a path for further in-depth analysis of those microbes by isolating and evaluating them for a valuable role in in vitro and in vivo studies.

## 1. Introduction

Baijiu is one of the most outstanding Chinese traditional liquors or white wine distilled from sorghum and wheat via a solid-state fermentation process. Baijiu has a unique history of 2000 years, existed as early as the Western Han Dynasty [1], and occupies a special status in traditional Chinese culture [2] Baijiu uses simple materials, grain (Zaopei), and water to produce complex flavor, and is available in several different varieties produced after four years of brewing buried in urns. The town of Moutai lies on the shore of the Chishui River in Guizhou Province of China and is famous for producing authentic liquor. Zaopei-flavored Baijiu represents one of the foremost flavored liquors in China. It has a unique aroma and lingering aftertaste and is enriched in thousands of volatile compounds.

Two main steps are involved in the mass production of branded Baijiu: the first step is the preparation of Daqu, and the second is fermentation and final production of liquor. Daqu (Appendix A) is a brick-shaped solid-state fermentation starter culture named for its large size [3]. The complex production process of Daqu leads to its higher microbial diversity and richer flavor than other simple starters, while less is known about its specific role and composition [3,4]. There are various fungal and bacterial species in Daqu. The bacterial effect is of great significance for producing organic acids responsible for developing liquor aroma [5]. Probiotics are viable bacterial or fungal species that confer health benefits when consumed or applied to the human body [6,7]. They can be found naturally in different sources, such as yogurt, fermented foods, and beverages, or added artificially in various food items, branded capsules, etc. Every fermented product, including food items, consists of a composite microbial community. This microbiota has indigenous microorganisms connected with the raw materials naturally and/or those existing on the apparatus/equipment or in the environment where the process is carried out, and on the surfaces of processing sites.

In contrast, chosen microorganisms may be used as starter cultures [8]. So, these microbes present in the starters are the players involved in fermentation and flavor development. Previously, numerous traditional methods were used to investigate the microbial communities and their essential role in liquor fermentation.

The conventional microbiological methods, i.e., isolation, culturing, identification, and PCR-DGGE (analysis), have identified specific microbes in alcohol fermentation (e.g., *Bacillus*, *Aspergillus*, *Weissella*, *Saccharomyces*, and *Lactobacillus*) [4,9,10]. In recent years, there has been less focus on the presence and investigation on microbes’ beneficial role in starter cultures. Recent advances in sequence-based techniques paved the way to providing in-depth metagenomics information of all culturable and non-culturable microbes in any environment. They support quick massive sequence data covering and revealing an extensive range of microbes and their functional roles [11,12], which may provide a bigger picture of microbes present in the starter culture [13]. This work may provide a road map to look for the beneficial role of those microbes, which can probably be used as probiotic strains after undergoing isolation, characterization, and validation in vitro and in vivo.

The most reliable sequencing method for identifying bacterial and fungal communities is using 16S/18S rDNA and ITS sequencing. These two methods are appropriate to identify the microbes present, while other related bioinformatics tools provide insight for associated functional analysis of those microbes. These methods investigate the microbiome diversity of various environments, including Chinese liquors, revealing the presence of complex microbes in those liquors and the secrets behind their traditional fermentation methods [14,15,16]. In the wine industry, microorganisms such as fungal and bacterial species are used in commercial starter cultures [17]. In most cases, lactic acid-producing bacteria are abundant in those starter cultures [18]. These complex microbial compositions are considered to contribute to the exceptional aromas and tastes of Baijiu [19].

Therefore, the present study aimed to (1) investigate the whole microbial diversity of the branded starters, (2) to identify the highly abundant microbial taxa, including fungi and bacteria, which may have a beneficial role based on literature mining, (3) to investigate the functional role of the microbes present in Baijiu Zaopei starter culture (Daqu). Uncovering those vital microbes and their functions in fermentation will improve understanding of the microbes’ beneficial role and possible future use of traditionally processed wine preparation techniques.

## 2. Materials and Methods

### 2.1. Sample Collection

Branded starters were collected from Renhuai Guizhou, China, with the assistance of Luzhou Ronghe Weizhen Biotechnology Co., Ltd. Luzhou, China followed by DNA extraction.

### 2.2. DNA Extraction and PCR Amplification

#### 2.2.1. Sample Collection

DNA was extracted using an Omega genuine D2561-02 Poly-Gel DNA Extraction Kit (50T) (Omega Bio-tek, Inc., Norcross, GA, USA) as per the manufacturer’s recommendation. Briefly, around 100 mg of dry Zaopei was ground with forceps manually and then added to a 1.5 mL grinding tube provided with the kit. DNA extraction from five (replicate) separate random sites from a brick of Daqu was carried out; DNA was quantified and stored at −80 ℃.

#### 2.2.2. Wet Sample

Around 50–100 gm of Zaopei (fermented grains, commonly named starter culture) was ground and added to a 20 mL tube. Later, autoclaved water was added and then mixed well to make a slurry. Further, the slurry was put in an oven at 37 °C for 4–5 h, and the DNA extraction was performed using manufacturer protocols of the Omega genuine D2561-02 Poly-Gel DNA Extraction Kit (50T) (Omega Bio-tek, Inc., Norcross, GA, USA). The quality and quantity of the isolated DNA were analyzed on a NanoDrop 2000 UV–Vis spectrophotometer (Thermo Scientific, Wilmington, DE, USA). The DNA samples were transported to Majorbio Bio-Pharm Technology Co., Ltd. (Shanghai, China) on dry ice. DNA extracted was subjected to 1% agarose gel, to see its purity and quality. For the bacterial community, bacterial primers 27F (5′-AGRGTTYGATYMTGGCTCAG-3′) and 1492R (5′-RGYTACCTTGTTACGACTT-3′) were used to amplify 16S bacterial rRNA genes [20]. For the fungal community, ITS1F (5′-CTTGGTCATTTAGAGGAAGTAA-3′) and ITS4R (5′-TCCTCCGCTTATTGATATGC-3′) primers were used to amplify ITS sequences [21]. Primers were tailed with PacBio barcode sequences to distinguish each sample. Amplification reactions of the 20 μL volume contained forward primer (5 μM) 0.8 μL, reverse primer (5 μM) 0.8 μL, 2.5 mM dNTPs 2 μL, 5× FastPfu buffer 4 μL, template DNA 10 ng, FastPfu DNA polymerase 0.4 μL, and DNase-free water. The PCR amplification conditions followed were denaturation at 95 ℃ for 3 min, followed by 27 cycles of denaturing at 95 °C for 30 s, later annealing at 60 °C for 30 s, and then the extension step at 72 °C for up to 45 s, a single extension step at 72 °C for 10 min, and the last step ended at 4 °C (ABI GeneAmp^®^ 9700 PCR thermocycler, San Diego County, CA, USA). All PCR reactions were performed in triplicate. After agarose gel electrophoresis, the obtained PCR products were cleaned and purified using the AMPure^®^ PB beads (Pacific Biosciences, San Diego County, CA, USA) and quantified with a specialized Quantus™ Fluorometer kit (Promega, WI, USA).

### 2.3. DNA Library Construction and Sequencing

Purified PCR products were then pooled in equimolar amounts, and a DNA library was created using the SMRTbell^®^ Express Template Prep Kit 2.0 (Pacific Biosciences, Menlo Park, CA, USA) as per PacBio’s instructions. The obtained Purified SMRTbell libraries were subjected to sequencing on the Pacbio Sequel II System (Pacific Biosciences, Menlo Park, CA, USA) by Majorbio Bio-Pharm Technology Co., Ltd. (Shanghai, China).

### 2.4. Data Processing

PacBio raw reads were primarily processed by using the SMRT Link software (version 8.0, Pacific Biosciences of California, Menlo Park, CA, USA) to acquire demultiplexed circular consensus sequence (CCS) reads. The criteria set for sequences was having a minimum of three full passes and 99% sequence accuracy. CCS reads were then barcode-identified and length-filtered. For the bacterial 16S rRNA gene, sequences with a length <1000 or >1800 bp were removed. For fungal ITS, sequences with a length of <300 bp or >900 bp were removed.

### 2.5. OTU Clustering Method

The optimized CCS reads were clustered into operational taxonomic units (OTUs) using UPARSE 7.1 [22,23] with a 97% sequence similarity level. The abundant sequence for each OTU was chosen as a characteristic sequence. To diminish the effects of sequencing depth on ecological parameters such as alpha and beta diversity, the number of 16S rRNA gene sequences from each sample were rarefied by up to 6000, which produced an average coverage of 99.09%.

The taxonomy of each OTU characteristic sequence was first analyzed using RDP Classifier version 2.2 [24] against the 16S rRNA gene database (eg. Silva v138), and later followed by a metagenomic function using Phylogenetic Investigation of Communities by reconstruction of Unobserved States (PICRUSt2) [25], which were all based on OTU representative sequences. PICRUSt2 is a package containing a sequence of tools: first of all, HMMER was cast off to align OTU sequences with a reference sequence. Later, EPA-NG and Gappa were utilized to put OTU sequences into a reference tree. The castor was used to normalize the 16S gene copies, and MinPath was used to predict gene family profiles further and locate the gene pathways. The entire analysis process was according to the protocols of PICRUSt2.

### 2.6. Statistical Analysis

Bioinformatic analysis was carried out using the Majorbio Cloud platform (https://cloud.majorbio.com). Based on the OTU information, the rarefaction curves and alpha diversity parameters, including Shannon index, Chao1 richness, observed OTU, etc., were calculated using Mothur v1.30.1 [26]. The similarity among the microbial communities in different samples was determined by principal coordinate analysis (PCoA) based on Bray–Curtis dissimilarity using the Vegan v2.5-3 package. The linear discriminant analysis (LDA) effect size (LEfSe) [27] (http://huttenhower.sph.harvard.edu/LEfSe) was used to identify the significantly abundant taxa (phylum to genera) of bacteria among the different groups (LDA score > 2 was used and *p* < 0.05 was considered significant).

## 3. Results

### 3.1. Overall Microbiome Composition of the Starter Culture

To oversee the overall microbiome diversity of the starter culture, we assessed the microbiome through high-throughput sequencing. In the bacterial diversity part, 142 OTUs were observed in all ten samples. The taxon distribution observed in starter culture consists of domain: 1, kingdom: 1, phylum: 6, class: 9, order: 32, family: 54, genus: 81, and the total number of species found was 125. In the fungal part, the taxa observed were distributed in the manner, with: domain: 1, kingdom: 1, phylum: 3, class: 8, order: 12, family: 27, genus: 44, species: 54, and total OTU: 65.

### 3.2. Alpha Diversity Indices in Wet and Dry Group

The alpha diversity parameters in the bacterial diversity portion, namely, Chao, Shannon, and Simpson, visibly differed in both groups. However, our focus was not on seeing the difference between dry and wet samples. In Table 1 and Table 2, it is shown that the number of OTUs increased in the wet group in each sample, which indicates that there were active culturable microbial taxa present both in the fungal and bacterial part, which started to replicate and grow when a comparatively suitable environment was provided.

### 3.3. Community Composition

As indicated in Figure 1A–C, in the case of bacteria, the highly abundant taxa on the phylum level and in both dry and wet samples were Firmicutes and Actinobacteria, followed by other unknown taxa. To see in depth, we analyzed the OTUs at the family and species level, indicating some interesting and beneficial taxa such as *Weissella cibaria*, *Weissella paramesenteroides*, *Pediococcus acidilactici*, *Lactobacillus curvatus*, and *Oceanobacillus caeni,* etc. (Figure 2A–C). Meanwhile, in the fungal part, the abundant taxa observed on the phylum level consist of highly abundant Ascomycota, followed by unknown taxa. Looking at species level diversity, it was observed that the highly abundant species in both the dry and wet samples were *Saccharomycopsis fibuligera*, *Wickerhamomyces anomalus,* and some other unknown, less abundant species.

### 3.4. Species Difference Analysis

Overall, the method is used to see various community-abundant species using multiple statistical methods in different groups. In our case, we did not see any significant difference in the overall composition of the taxa. Still, our results demonstrate that based on the Wilcoxon rank abundance test, the highly abundant bacterial taxa on the species level were *Weiselia ciberia*, *Weissella paramesenteroides*, *Lactobacillus curvatus*, *Oceanobacillus caeni,* and some other less abundant species (Figure 3A and Appendix A). In the fungal part, the highly abundant species based on the Wilcoxon sum test were *Mucor racemosus* and *Microsasus brevicaulis*, followed by *Issatchenkia oriantels,* and some less abundant species (Figure 3B). In analysis through *LEfSe* both in a dry and wet group, the genus *Weissella* was dominant and followed by other genera such as *Lactobacillus* and *Bacillus* (Figure 3C and Appendix A). In the case of fungi, the highest LDA score was obtained by *Mucor racemosus* and *Basidiomycota* in the dry group and *Lichtheimiaceae,* unclassified *Lichtheimiaceae*, and *issatchenkia* (Figure 3D).

### 3.5. Functional Predictive Analysis

PICRUSt is a software package for the functional prediction of 16S amplicon sequencing results, which can perform functional predictive analysis on 16S, 18S, or ITS sequencing data. Databases such as Orthologous Groups of Proteins (COGs), which attempted a phylogenetic classification of the proteins encoded in 21 complete genomes of bacteria, archaea, and eukaryotes, the Kyoto Encyclopedia of Genes and Genomes database (KEGG, http://www.genome.jp/kegg/), Tax4Fun Feature Prediction, and Fungi Functional Guild (FUNGuild) were used. Unique COG and KO abundance pathways and enzymes were linked in taxon-associated dry and wet groups. In the bacterial diversity section, both dry and wet groups, through KEGG analysis, the highly abundant associated pathways found were carbohydrate metabolism, amino acid metabolism, membrane transport, and energy metabolism, followed by various others as shown in the figures and Appendix A (Figure 4A,B and Appendix A). In addition, Bug Base phenotype prediction identified Gram-positive, Gram-negative, biofilm-forming, pathogenic, mobile element-containing phenotype types (Appendix A), and FAPROTAX function prediction revealed nitrification, denitrification, etc. (Appendix A).

## 4. Discussion

The presence and role of beneficial microbial communities in the starters have not yet been thoroughly investigated. Previous studies have investigated Moutai or related liquor for overall microbial diversity and functional analysis [14,28]. There has been no such study where the investigation focused on the presence of beneficial microbes in liquor, where mainly we have focused on environment-related branded liquor starters. Previous studies have demonstrated the role of *Weisslia cibaria*, *Weissella paramesenteroides*, *Pediococcus acidilactici*, and *Lactobacillus curvatus* probiotic functions [28,29,30,31,32], which provides a base for our current study focus on the identification of the beneficial microbes in environment-related branded liquor starter. The present study identified various bacterial and fungal species which are beneficial. The most dominant taxa found in the current study were *Weissella cibaria*, followed by less abundant *Weissella paramesenteroides*, *Pediococcus acidilactici*, *Lactobacillus curvatus*, *Saccharomycopsis fibuligera*, and *Wickerhamomyces anomalus,* respectively. Studies such as one recently published by Huang et al. demonstrated that *Weissella cibaria* isolated from Chinese sauerkraut attenuates gut barrier-related protein and inflammation, and influences the signaling pathway in a gut cell line; proving this probiotic strain’s role [33]. *Weissella paramesenteroides* is considered as a potential probiotic strain and was evaluated recently in various in vitro and in vivo studies [34]. *Lactobacillus curvatus* has been isolated from multiple sources, and its probiotic role and efficiency have been well documented previously [35].

Similarly, it has been reported that *P. acidilactici* is generally considered safe and possesses probiotic properties such as beneficial enzymatic activity [36]. It is widely used in food fermentation and as a starter for cheese and yogurt production [37]. Therefore, the results of our study suggest and give a clue that *Weissella paramesenteroides*, *Pediococcus acidilactici*, and *Lactobacillus curvatus* could have great potential for future development as probiotic strains.

The current study observed a total of 142 OTUs in the bacterial diversity part and a total of 65 OTUs in the fungal diversity part. The total number of OTUs varies in various studies, and the microbial diversity composition also varies. This could be associated with variations in processing method, environmental factors such as temperature, environmental microbial load, and source of raw material used in the starter culture [38,39]. As in our case, we directly obtained DNA from the starter raw composite called Daqu and followed this up with a study of microbial diversity, so there is a difference in microbial diversity compared with other studies. We believed this might also differ from all microbes and their metabolites and by-products taking part in fermentation and developing liquor flavor [40,41].

Alpha diversity refers to the variety within a particular area or ecosystem. It is usually expressed by the number of species (i.e., species richness) in that ecosystem [42]. In the bacterial diversity portion, the alpha diversity parameters, namely, Chao, Shannon, and Simpson, visibly differed in both groups, and the number of OTUs increased in the wet group in each sample, which indicated that there are active culturable microbial taxa present both in fungal and in bacterial parts, which might have started to replicate and grow when a comparatively suitable environment was provided, as reported by [43]. In the case of bacteria, the highly abundant taxa on the phylum level in both dry samples were Firmicutes and Actinobacteria, followed by other unknown taxa, similar to what has been reported previously [44]. A similar type of predominant observation of Firmicutes and Actinobacteria in Maotai, a similar brand, was used as a microflora starter in liquor Maotai synthesis reported in previous related studies [44,45], while another report showed that Daqu samples contained Proteobacteria together with Firmicutes and Actinobacteriota [45].

Actinobacteria are mostly saprophytic, soil-dwelling organisms and contribute significantly to the conversion and degradation of complex biopolymers, such as lignocellulose, hemicellulose, pectin, keratin, and chitin [46], which shows their potential in nutrient cycles. Furthermore, it is demonstrated that their exoenzymes may help facilitate feed utilization and digestion once they colonize the host intestine. In addition, the colonizing microflora plays an essential role in the resistance to infectious diseases by producing antibacterial substances, which is one of the probiotics’ key characteristics [47]. Moreover, as Wang et al. [48] suggested, adhesion and colonization of the probiotic strain on the mucosal surfaces are possible protective mechanisms against pathogens. The protection occurs through competition for binding sites and nutrients [49] or immune modulation and is a prominent feature of probiotics [50]. These characteristics are also anticipated from Actinobacteria through the production of secondary metabolites. Actinomycetes are well studied for secondary metabolites; however, well studied taxa have the potential to yield new metabolites because of unanticipated biosynthetic gene clusters [51].

The previous literature reported that bacteria, especially high-temperature bacteria, are predominant following molds and yeasts in white wines, especially Daqu [52]. It is well known that during the Daqu-making process, with the temperature increasing, some high-temperature bacteria, such as Bacillus, should gradually become the predominant bacteria [53]. However, in this study, Bacillus was not the dominant bacterial strain as the sampling was before the high heat stage. So, it contained almost all microbial strains and was dominated by various strains, and most of them have been proven to have a beneficial role [28,36]. Previous studies have reported that high-temperature bacteria can produce all kinds of hydrolytic enzymes (such as amylase and protease) and many flavor components (including alcohols, aldehydes, acids, esters, pyrazines, and aromatic compounds) [54], as well as some enzymes and aromas or other functions in the wine industry. It is known that the metabolic products of bacteria are essential in developing unique flavors for liquor [54,55]. The current, in-depth study identified some exciting and beneficial taxa such as *Weissella cibaria*, *Weissella paramesenteroides*, *Pediococcus acidilactici*, *Lactobacillus curvatus*, and *Oceanobacillus caeni,* which are quite in accordance with the previous studies [56]. The microbiota as mentioned above with a lot of other lactic acid bacterial strains was also reported from other similar types of studies and they may add up in developing flavor in liquor [56,57], while probiotics and the beneficial role of those strains as mentioned above are not a hidden mystery.

The fungal part of the abundant taxa observed on the phylum level consisted of highly abundant Ascomycota, followed by unknown taxa. Looking at species level diversity, it was observed that the highly abundant species in both the dry and wet samples were *Saccharomycopsis fibuligera*, *Wickerhamomyces anomalus*, and some other unknown, less abundant species. These fungi’s unique roles in fermentation, flavor development, and, most specifically, probiotics have been well documented previously [58,59,60]. Note that the presence of abundant *Saccharomycopsis fibuligera* in Figure 2 and, in contrast, others as the main fungi in Figure 3 is because of different analyses and focus based on either looking for abundance or comparing various species in both the groups and methods used. Further in-depth functional prediction by aligning the amino acid sequences in KEGG, PICRUSt, COGs, and Tax4Fun revealed unique innovative pathways linked with carbohydrate metabolism, amino acid metabolism, membrane transport, and energy metabolism. The three most dominant COG functions are RNA processing and modification, chromatin structure, dynamics, energy production and conversion, etc. Although there are limited related studies related to the functional prediction of the microbiome of Baijiu or related wines, one consistent study reported contrasting results [27].

## 5. Conclusions

In this study, we revealed the microbial composition of branded starter samples and the functional gene composition of the starter microbiome. This study identified the presence of beneficial microbes in the starter culture. It opened a path for further in-depth analysis of those microbes by isolating them and evaluating their beneficial role in vitro and in vivo. Specifically, the study identified *Weissella cibaria*, a bacterial strain, and *Saccharomycopsis fibuligera* as the prominent species present in all the samples. Previous studies have demonstrated the strong beneficial role of *Weissella cibaria*, which provides a base for focusing on further characterization of our strain to be further processed for its probiotic activity through isolation, whole genome sequencing, and follow-up in vitro and in vivo investigation.

## Figures and Tables

**Figure 1 foods-12-00025-f001:**
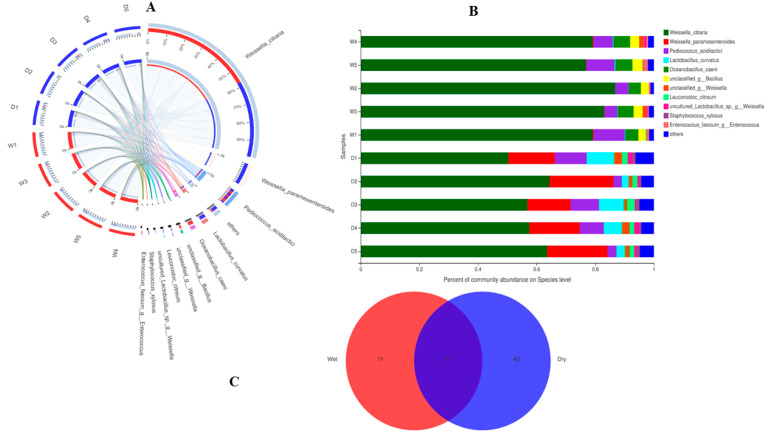
(**A**) (Bacteria) In the Circos sample and species relationship diagram, the small semi-circle (left half circle) represents the composition of species in the sample, the color of the outer ribbon represents which group it comes from, the color of the inner ribbon represents the species, and the length represents the species’ relative abundance in the corresponding sample; the large semicircle (right half circle) represents the distribution ratio of species in different samples at the taxonomic level, the outer ribbon represents species, the inner ribbon color represents different groups, and the length represents the species’ proportion of distribution in a sample. (**B**) (Bacteria) The abscissa/ordinate is the sample name, the ordinate/abscissa is the proportion of species in the sample, the columns of different colors represent different species, and the column length represents the proportion of the species. (**C**) (Bacteria) The graph shows a Venn diagram, different colors represent different groups (or samples), the numbers in the overlapping parts represent the number of species shared by multiple groups, and the numbers in the non-overlapping parts represent individual species number in a group.

**Figure 2 foods-12-00025-f002:**
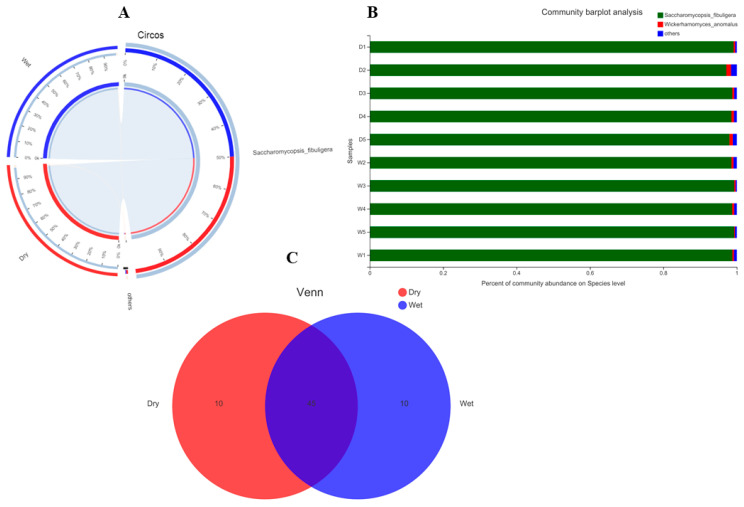
(**A**) (Fungi) In the Circos sample and species relationship diagram, the small semicircle (left half circle) represents the composition of species in the sample, the color of the outer ribbon represents which group it comes from, the color of the inner ribbon represents the species, and the length represents the species’ relative abundance in the corresponding sample; the large semicircle (right half circle) represents the distribution ratio of species in different samples at the taxonomic level, the outer ribbon represents species, the inner ribbon color represents different groups, and the length represents the species’ proportion of distribution in a sample. (**B**) The abscissa/ordinate is the sample name, the ordinate/abscissa is the proportion of species in the sample, the columns of different colors represent different species, and the column length represents the proportion of the species. (**C**) (Fungi). The graph shows a Venn diagram, different colors represent different groups (or samples), the numbers in the overlapping parts represent the number of species shared by multiple groups, and the numbers in the non-overlapping parts represent the number of species unique to the corresponding group if the group is greater than or equal to 6.

**Figure 3 foods-12-00025-f003:**
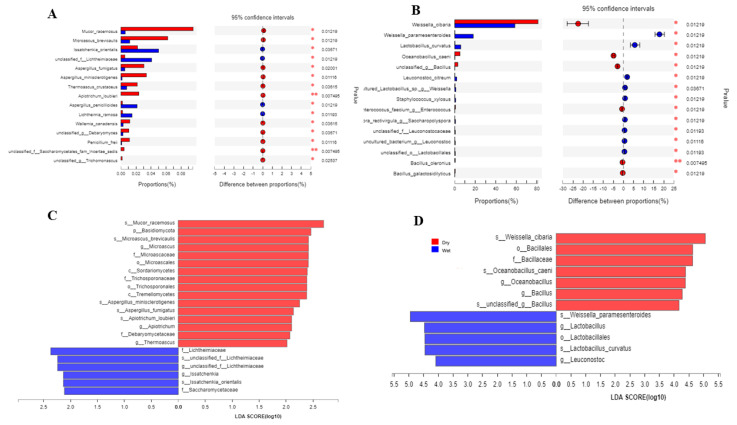
Colors in the figure show different groups. Names of specific taxa are displayed on the left; highly abundant tax are on the top, and the corresponding *p* value on the right represents a difference in the presence of that particular taxon in the groups. In contrast, the middle dots represent the confidence interval. On the right, (**B**), are bacterial species, and on the left, (**A**), are taxon differences of fungal species. (**C**) Comparison and analysis of the relative differential abundance of fungi based on LDA score. (**D**) Comparison and analysis of the relative differential abundance of bacteria based on LDA score. LDA score table showing unique abundant taxa in each group of fungi and bacteria which gained LDA score ≥ 2.

**Figure 4 foods-12-00025-f004:**
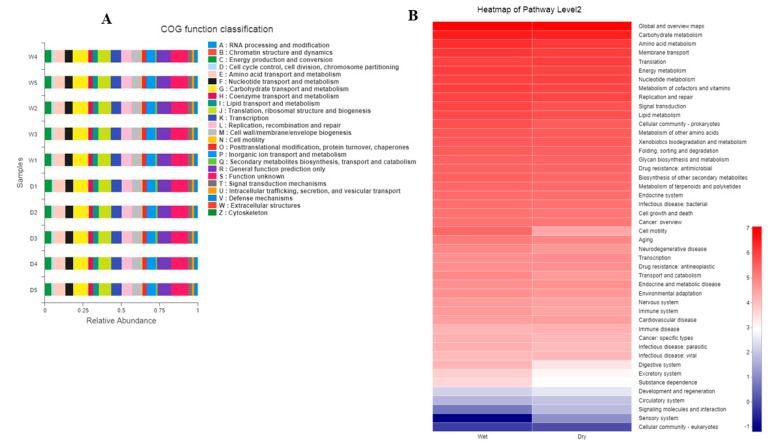
(**A**) CO Functional Classification; (**B**) KEGG Heatmap.

**Table 1 foods-12-00025-t001:** Bacterial Alpha Diversity Table (OTUs).

Sample	Shannon	Simpson	Ace	Chao	Coverage
D1	1.785958	0.300804	121.0517	114	0.999425
D2	1.265642	0.464346	98.78463	99	0.999604
D3	1.596225	0.36266	120.618	116.8333	0.999683
D4	1.546949	0.372019	104.6294	108.125	0.999555
D5	1.347136	0.449613	106.2161	102.4615	0.999639
W1	0.848414	0.641035	64.65278	67.16667	0.999732
W2	0.612391	0.758397	45.38116	43.66667	0.999914
W3	0.803193	0.695663	73.43487	67.23077	0.999771
W4	0.946436	0.635935	88.74204	84.90909	0.999625
W5	0.962557	0.604723	94.97036	89.92857	0.999594

**Table 2 foods-12-00025-t002:** Fungal Alpha Diversity Table (ITS).

Sample ID	Shannon	Simpson	Ace	Chao	Coverage
D1	0.07202	0.98269	47.7492	47	0.99984
D2	0.20213	0.94355	51.7504	52.2	0.99991
D3	0.09726	0.97601	40.2285	37.625	0.99982
D4	0.11512	0.97067	44.8752	44.6	0.99981
D5	0.14936	0.96059	53.3065	49.625	0.99976
W1	0.09453	0.97641	25.8509	26	0.99996
W2	0.11303	0.97142	45.6741	43.625	0.99986
W3	0.05957	0.9856	36.3077	36.5	0.99992
W4	0.09984	0.97501	42.1254	40.875	0.99988
W5	0.06084	0.98583	27.2846	27	0.99998

## Data Availability

Data are contained within the article.

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
