# Peer review of "Metagenomic Analysis of Liquor Starter Culture Revealed Beneficial Microbes’ Presence"

_foods, 2022, doi:10.3390/foods12010025_

Round 1

Reviewer 1 Report

In this manuscript authors present the methods for characterisation of microbial ecosystem of Zaopei Baijiu Daqu which is starter culture for Baijiu, Chinese white wine. So far, less was known about Daqu specific role and composition. It contains various fungal and bacterial species, also probiotics, which play special role in fermentation and flavor development. Traditional microbiological methods have identified only some specific microbes in alcohol fermentation. Recently, advance in sequence-based techniques provides in-depth metagenomic information of all microbes in any environment. Thus, the main purpose of this study is to identify the vital microbes and their functions in fermentation. 

The title completely reflects the considered area.

The manuscript is clear written, the well-defined structure was followed, and it is relevant to the field of Food Biotechnology.

Literature review is extensive, most of the cited references are recent. Of course, some of them are older due to description of conventional microbiological methods and comparison of some results.

I assume that in Materials and methods section the experimental design is described enough in detail, so that the study could be repeated by any other researcher. References for all software items are also included.

My only concern is regarding the tables and figures. In both tables the units are missing in which Alpha diversity parameters (Shannon, Simpson....) are expressed. Probably OTU units are used. In any case, this should be noted.

Furthermore, from existing figures it is almost impossible to read any text. Without the explanation in subchapter “Community Composition” the reader would not know which are the most abundant taxon on phylum level in the case of bacteria and in the fungi part. Letters should be bigger, or authors should consider an option to divide original Figure 1(and others) into 3 parts.

Otherwise, the figures and tables properly show the data, and they can be understand.

Statistical analysis was performed, and corresponding software is listed.

I miss Conclusion section as a separate part of the manuscript. In the present version authors describe the conclusions of their research work in the last paragraph of Discussion section, which is possible. Nevertheless, with an independent Conclusion section, the main findings would be more emphasized

  •  

Author Response

Reviewer 1

  1. My only concern is regarding the tables and figures. In both tables, the units missing in which Alpha diversity parameters (Shannon, Simpson....) are expressed. Probably OTU units are used. In any case, this should be noted.

Answer: Thank you for the reviewer’s suggestion and recommendation; the word OTU and ITS have been added to both tables

  1. Furthermore, from existing figures it is almost impossible to read any text. Without the explanation in subchapter “Community Composition” the reader would not know which are the most abundant taxon on phylum level in the case of bacteria and in the fungi part. Letters should be bigger, or authors should consider an option to divide original Figure 1(and others) into 3 parts. Otherwise, the figures and tables properly show the data, and they can be understand.

Answer: Thank you for pointing out this critical issue in the figures. In the first submission, figures were provided with the text, and the file contains embedded figures (leading to low resolution) in a word file and so not visible; I will provide high-resolution TIF figures to be visible or will split figures if we're still not visible as advised by the reviewer.

  1. I miss Conclusion section as a separate part of the manuscript. In the present version authors describe the conclusions of their research work in the last paragraph of Discussion section, which is possible. Nevertheless, with an independent Conclusion section, the main findings would be more emphasized. 

Answer: conclusion section has been separated with a separate tile from the discussion in the new version.  

Reviewer 2 Report

Dear Authors,

your research manuscript concerns metagenomic analysis of special starter culture for production of white wine in China. You  sequenced the bacterial 16S rDNA gene and the fungal ITS region and determined the microbiome of the tested starter culture. You used various bioinformatics software to analyze data on the taxonomic diversity of microorganisms, their abundance, and assessed the potential metabolic pathways and phenotypic features of the most abundant taxa in the context of potential probiotics. The manuscript is very valuable and increases the knowledge about the starter microbiome and shows the potential application of many bioinformatics and statistical tools. The English language also needs correction.

Detailed comments:

Title – It should be “Metagenomic analysis…”

Abstract – You should add “and ITS” in L19 writing about yeast diversity.

Introduction – This part of manuscript is quite well written, I suggest adding some other methods used for genetic analysis besides PCR-DGGE (L60).

Materials and methods – in point 1.2.1 the word “collection” should be deleted (L97), because you wrote about extraction of DNA from dry samples in opposite to wet samples (1.2.2.). The word "almost" is not needed in L101.

Results  - Under tables 1 and 2 (P5), you should add explanations of the abbreviations D and W.

Throughout the results section, the current name of the bacterium  Latilactobacillus curvatus should be provided next to the previous name (Lactobacillus curvatus) so that the reader is aware of the name change.

In L197 figure 2A should be changed to 2A-C and moved to the next sentence concerning fungi.

In the "Species difference analysis" section, some names of microorganisms should be corrected (italic, lowercase species names, uppercase generic names).

L257 base name: Kyoto Encyclopedia.... is repeated.

 I think that in L267 the drawing of S2 should be corrected for S3. In addition, nowhere do the authors refer to Figures S4 and S5.

Figures 1-4 are not legible enough, the font size of the names of microorganisms should be slightly larger. Captions to the figures should include information whether they describe bacteria or fungi, because you have to guess. Fig. 1 and 2 - point C "...parts represent", but it is not known what.

The caption to Figure 3 should be corrected because parts A and B are interchanged and there is no explanation for Figure D.

I ask  to explain why Saccharomycopsis and Wickerhamomyces appear as the main fungi at Figure 2 and Mucor and other fungi at Figure 3. Perhaps the description in the text is insufficient.

Discussion – quite well written, but you should check and correct the names of microorganisms as in Results section.

References - You should adapt the references to the requirements of the journal. Some publications are quite old, it may be worth changing them to newer ones or removing them, e.g. ref. 19 ("current taxonomy" of lactic acid bacteria from 1997 is not current now).

Author Response

  1. The English language also needs correction.

Answer: Corrections have been made, and the whole manuscript has been edited for English mistakes. Changes can be found in the track changes form in the file. 

  1. Title – It should be “Metagenomic analysis…”

Answer: Corrections have been made.

  1. Abstract – You should add “and ITS” in L19 writing about yeast diversity.

Answer: Corrected as directed.

  1. Introduction – This part of the manuscript is quite well written; I suggest adding other methods used for genetic analysis besides PCR-DGGE (L60).

Answer: It’s a valid point to add more methods, such as temperature gradient gel electrophoresis (TGGE), emulsion PCR (ePCR), and droplet digital PCR (ddPCR), if the sentence was referred to general microbial analysis. Here in this sentence, we refer to methods previously used for microbe identification involved in alcohol fermentation, which, to the best of our knowledge based on the literature we found, are those mentioned and reference provided.   

  1. Materials and methods – in point 1.2.1 the word “collection” should be deleted (L97), because you wrote about the extraction of DNA from dry samples in opposition to wet samples (1.2.2.). The word "almost" is not needed in L101.

Answer: Thanks for the step-by-step and to-the-point input in improving the article, corrections have been made accordingly.

  1. Results - Under tables 1 and 2 (P5), you should add explanations of the abbreviations D and W.

Answer: Once again appreciate the time taken by the reviewer to point all those major and minor mistakes. Correction have been made. 

  1. Throughout the results section, the current name of the bacterium  Latilactobacillus curvatus should be provided next to the previous name (Lactobacillus curvatus) so that the reader is aware of the name change.

Answer: Not able to find this suggestion in the manuscript the reviewer is pointing

  1. In L197, figure 2A should be changed to 2A-C and moved to the next sentence concerning fungi.

Answer: Figure names have been corrected as advised,

  1. In the "Species difference analysis" section, some names of microorganisms should be corrected (italic, lowercase species names, uppercase generic names).

Answer: The concern about names is valid, and names should be as advised, but it’s a bit more time-consuming process. As mentioned in the article, we also used a Major bio cloud website-based platform for the analysis. Figures are generated through the database; I have asked the service provider for modification. It might take a long time to fix and get new figures. We will try our best to include the corrected figure in our next version.

  1. L257 base name: Kyoto Encyclopedia.... is repeated.

Answer: Corrections have been made

  1. I think that in L267 the drawing of S2 should be corrected for S3. In addition, nowhere do the authors refer to Figures S4 and S5.

Answer: Corrected have been made, and figures not refereed/mentioned in the text have been mentioned

  1. Figures 1-4 are not legible enough, the font size of the names of microorganisms should be slightly larger. Captions to the figures should include information whether they describe bacteria or fungi, because you have to guess. Fig. 1 and 2 - point C "...parts represent", but it is not known what.

Answer: Figure resolution has issue and we will ask the editor to add the high-resolution figures or will split the figures to make it able to visible and able to read. Further, the word fungi or Bacteria have been added in figure captions to make it easy to spot if the figure is about fungi or bacteria.

  1. The caption to Figure 3 should be corrected because parts A and B are interchanged and there is no explanation for Figure D.

Answer: We are thankful for the reviewer's effort and fruitful suggestion and recommendation. The correction has been made in the figure captions, and information has been added to figure D as well.  

  1. I ask to explain why Saccharomycopsis and Wickerhamomyces appear as the main fungi at Figure 2 and Mucor and other fungi at Figure 3. Perhaps the description in the text is insufficient.

Answer: The analysis in figure 1 is based on abundance-based which is our main target in this study. While the figure and taxon shown in figure 3 are based on a comparison of taxon differences in between the dry and wet groups. A short description has been added in the discussion section.

  1. Discussion – quite well written, but you should check and correct the names of microorganisms as in the Results section.

Answer: Article have been checked for name correction in the result and discussion section.

  1. References - You should adapt the references to the requirements of the journal. Some publications are quite old, it may be worth changing them to newer ones or removing them, e.g. ref. 19 ("current taxonomy" of lactic acid bacteria from 1997 is not current now).

Answer: Corrections have been made, and Reference have been updated.